# The Effect of the PhoP/PhoQ System on the Regulation of Multi-Stress Adaptation Induced by Acid Stress in *Salmonella* Typhimurium

**DOI:** 10.3390/foods13101533

**Published:** 2024-05-15

**Authors:** Xu Gao, Jina Han, Lixian Zhu, George-John E. Nychas, Yanwei Mao, Xiaoyin Yang, Yunge Liu, Xueqing Jiang, Yimin Zhang, Pengcheng Dong

**Affiliations:** 1Lab of Beef Processing and Quality Control, Shandong Agricultural University, Taian 271018, China; gaoxu@meatsci.cn (X.G.); zhlx@sdau.edu.cn (L.Z.); gjn@aua.gr (G.-J.E.N.); maoyanwei@163.com (Y.M.); yangxiaoyin@sdau.edu.cn (X.Y.); liuyunge@sdau.edu.cn (Y.L.); jxq@pcdong.cn (X.J.); ymzhang@sdau.edu.cn (Y.Z.); 2International Joint Research Lab (China and Greece) of Digital Transformation as an Enabler for Food Safety and Sustainability, Taian 271018, China; 3Shandong Provincial Key Laboratory of Poultry Diseases Diagnosis and Immunology, Poultry Breeding Engineering Technology Center of Shandong Province, Poultry Institute, Shandong Academy of Agricultural Sciences, Jinan 250023, China; hanjina2013@163.com; 4Department of Food Science and Human Nutrition, Agricultural University of Athens, 11855 Athens, Greece

**Keywords:** *Salmonella typhimurium*, two-component regulatory system, acid tolerance response, cross-protection, cationic antimicrobial peptide resistance mechanisms

## Abstract

Acidic stress in beef cattle slaughtering abattoirs can induce the acid adaptation response of in-plant contaminated *Salmonella*. This may further lead to multiple resistance responses threatening public health. Therefore, the acid, heat, osmotic and antibiotic resistances of *Salmonella typhimurium* (ATCC14028) were evaluated after a 90 min adaption in a pH = 5.4 “mild acid” Luria–Bertani medium. Differences in such resistances were also determined between the ∆*phoP* mutant and wild-type *Salmonella* strains to confirm the contribution of the PhoP/PhoQ system. The transcriptomic differences between the acid-adapted and ∆*phoP* strain were compared to explore the role of the PhoP/Q two-component system in regulating multi-stress resistance. Acid adaptation was found to increase the viability of *Salmonella* to lethal acid, heat and hyperosmotic treatments. In particular, acid adaptation significantly increased the resistance of *Salmonella typhimurium* to Polymyxin B, and such resistance can last for 21 days when the adapted strain was stored in meat extract medium at 4 °C. Transcriptomics analysis revealed 178 up-regulated and 274 down-regulated genes in the ∆*phoP* strain. The *Salmonella* infection, cationic antimicrobial peptide (CAMP) resistance, quorum sensing and two-component system pathways were down-regulated, while the bacterial tricarboxylic acid cycle pathways were up-regulated. Transcriptomics and RT-qPCR analyses revealed that the deletion of the *phoP* gene resulted in the down-regulation of the expression of genes related to lipid A modification and efflux pumps. These changes in the gene expression result in the change in net negative charge and the mobility of the cell membrane, resulting in enhanced CAMP resistance. The confirmation of multiple stress resistance under acid adaptation and the transcriptomic study in the current study may provide valuable information for the control of multiple stress resistance and meat safety.

## 1. Introduction

*Salmonella* is a common foodborne bacterium that causes gastrointestinal illness and is considered to be the second most prevalent foodborne bacterium in humans, according to the EFSA Surveillance Report on Recognition of Co-occurring Diseases 2021 [1]. Although various strategies have been implemented either to control contamination in various stages of beef production, i.e., farming, slaughter, splitting and transportation, beef is still a significant source (5.7–9.15%) of *Salmonella* outbreaks [2]. Among the various measures taken to control the microbial population on beef, the use of the high-pressure spraying of water with 2–5% lactic acid at 55 °C is the most common practice in most countries [3]. However, the leftover lactic acid causes the production of wastewater as well as a possible decrease locally in post-slaughter carcasses in pH (reaching an ultimate pH of 5.4), which can result in a mildly acidic pH environment (pH 4.5–6). This rather acidic environment could stimulate the development of acid stress protection in pathogens, i.e., to induce acid tolerance response (ATR) [4,5,6,7]. It needs to be noted that cross-protection against other factors such as high temperature, hypertonicity and oxidation was also revealed [8,9]. On the other hand, acid-adapted cells of *Salmonella*, *C. sakazakii*, *L. monocytogenes* and *A. baumannii* found to decrease their sensitivity against common antibiotics, i.e., tetracycline, chloramphenicol, ciprofloxacin and sulfonamides [10,11,12,13]. Cationic antimicrobial peptides (CAMPs) are considered the next generation of bacteriostatic substances discovered after antibiotics [14] since their active target is the cell membrane, minimizing the possibility of the resistance development of bacterial cells and an indirect reduction in virulence [15,16]. This very promising view has been under dispute by Perez and Groisman [17] who found a 100,000-fold increase in *Salmonella* survival under the CAMP polymyxin B after acid adaptation at pH 5.8. Furthermore, recent transcriptomic studies have shown that the CAMP resistance pathway of *Salmonella* is significantly up-regulated under mild acid acclimation [18].

It is well known that bacteria have developed communication mechanisms that allow them to understand changes in the environment and therefore to leverage specific strategies that allow them to survive in space and time within their niche [19]. One of the essential strategies in the sensing and signaling mechanisms between bacteria and the environment is the two-component system (TCS), which plays a key role in how changes in the external environment are perceived by the bacteria and how quickly they respond to the changes [20]. The TCS generally consists of sensor kinases (SKs) in the periplasm and response regulators (RRs) in the cytoplasm [21]. When the SK is activated by a changing environment, it self-phosphorylates and delivers the phosphate group to a homologous RR. The RR that receives the signal will bind to specific downstream promoters to regulate gene expression and complete the process of cellular adaptation to environmental changes [22].

*Salmonella* has been identified with several environmental sensing-related TCSs such as BaeR/S (envelope, bile and multidrug stress), CpxR/A (heat, AMP, envelope stress), OmpR-EnvZ (acid and osmotic stress), RcsBCD (envelope and AMP stress) and PmrA/B (AMP stress). Acid adaptation is related to PhoP/Q, while *phoQ* is activated by externally moderately acidic H^+^, low Mg^2+^ and cationic antimicrobial peptides. Then, *phoP* regulates virulence, stress resistance and the combined effects of other TCSs [19,23,24]. Although the correlation between PhoP/Q and various kinds of resistance has been widely studied, there is still a lack of valid information regarding its effects on various potential adaptations of bacteria cells and its regulatory fitness mechanisms from a global perspective. The development of transcriptomics and other related techniques could shed light on this matter.

Therefore, this study aimed to investigate the effect of acid adaptation and PhoP/Q on the multi-tolerance (acid, heat, salt and antibiotics include the considered new generation of antibiotic CAMPs) of *Salmonella*. The survivability of acid adaptation and gene deletion to multiple environmental resistances in *Salmonella* was determined. Transcriptomics and RT-qPCR analyses were then further performed to decipher the global regulator aspect of PhoP/Q and to explore possible mechanisms by which this TCS plays its role in stress tolerance.

## 2. Materials and Methods

### 2.1. Bacterial Strains

The strain used in this study was the wild-type strain (WT) of *Salmonella typhimurium* (ATCC 14028). The *phoP* gene deletion mutation strain (∆*phoP*) was constructed using *Salmonella typhimurium* and kept in the Animal Product Processing Laboratory of Shandong Agricultural University [25]. All the strains were stored in non-glucose tryptic soy broth (NTSB, Beijing Land Bridge Technology, Beijing, China) supplemented with 25% (*v*/*v*) glycerol (Tianjin Kaitong Chemical Reagent, Tianjin, China) at −80 °C.

### 2.2. Acid Adaptation of WT and ∆phoP Strains

The strains (WT and ∆*phoP*) were inoculated at 1‰ inoculum in Luria–Bertani (LB) broth (Beijing Land Bridge Technology, Beijing, China) for 18 h (37 °C, 200 rpm) to activate the strain, and the strain that was activated will be activated a second time under the same condition. Two full loops of the twice-activated WT and ∆*phoP* strains were incubated in a pH = 5.4 LB broth (adjusted by lactic acid, Tianjin Kaitong Chemical Reagent, Tianjin, China) for 90 min at 37 °C to obtain the acid-adapted strain. At the same time, another two full loops of the twice-activated WT and ∆*phoP* strains were incubated in a pH = 7 LB broth (adjusted by phosphates) for 90 min at 37 °C to obtain the non-adapted group [18]. The non-adapted WT strain was set as the control.

### 2.3. Determination of Acid, Heat and Salt Stress Tolerance of Adapted/Non-Adapted WT/Δphop Strains

For the acid tolerance experiment, the strains obtained in Section 2.2 (non-adapted WT, acid-adapted WT, non-adapted ∆*phoP* and acid-adapted ∆*phoP*) were diluted with peptone saline (0.1% petone, Beijing Land Bridge Technology, Beijing, China and 0.85% NaCl, Sinopharm Chemical Reagent, Shanghai, China) and finally resuspended in LB at pH 3 (pH adjusted with hydrochloric acid, Sinopharm Chemical Reagent, Shanghai, China), with an initial concentration of 10^6^ CFU/mL. The non-adapted WT strain was set as the control. The acid stimulation experiment was carried out for 2 h, during which 1 mL of the bacterial solution was aspirated for dilution every 30 min and then subjected to a colony counting experiment on LB agar medium [25]. A linear regression equation of log 10 values of colony counting results and time was fitted using Microsoft Office Excel. *D*-values were calculated from the negative inverse values of their slopes. The experiment was repeated three times independently, and the average results were taken.

The heat stress tolerance tests were prepared as previously described [26] with some modifications to the culture media. The initial bacterial concentration for the heat-resistant line assay was 10^6^ CFU/mL; 1.5 mL of the bacterial solution (prepared in Section 2.2) was transferred separately into centrifuge tubes and placed in a 55 °C water bath. The centrifuge tubes were removed at 0, 5, 10, 15, 20, 25 and 30 min for an immediate cold water bath and then subjected to a colony counting experiment on LB agar medium. *D*-values consistent with acid tolerance experiments were used to indicate the survival of strains in high-temperature environments.

For the salt tolerance experiment, the strain obtained in Section 2.2 was diluted with peptone saline and then resuspended in LB containing 8% *w*/*v* NaCl, with an initial concentration of 10^6^ CFU/mL. Then, it was incubated in the high-salt environment for 4 days, and at 0, 1, 2, 3 and 4 days, 1 mL of dilution was taken to the appropriate gradient on LB agar and then counted, respectively. Logarithmic reduction was counted to indicate the ability of strains to survive in hyperosmotic environments.

### 2.4. Determination of Antibiotic Resistance of Adapted/Non-Adapted WT/Δphop Strains

The minimum inhibitory concentration (MIC) was determined by the broth microdilution method as recommended by CLSI M100-33. Briefly, an antibiotics stock solution was prepared (polymyxin B, ceftazidime, Gentamycin, ampicillin) at a stock concentration of 5120 µg/mL. And then, the antibiotics stock solutions were diluted into Mueller–Hinton broth (Qingdao Hi-tech Industrial Park Hope Bio-technology, Qingdao, China) at concentrations of 0.25, 0.5, 1, 2, 4, 8, 16, 32, 64, 128, 256 and 512 µg/mL, respectively. Less than 5% Dimethyl sulfoxide (DMSO, Beijing Solarbio Science & Technology, Beijing, China) was added as a co-solvent to ampicillin that is not readily soluble in water (this concentration has been shown to have no bacteriostatic effect) [27]. The strain obtained in Section 2.2 was adjusted to a 10^6^ CFU/mL initial concentration. A total of 100 uL of bacterial solution and different concentrations of antibiotic solution were added to a 96-well plate, respectively; six repeat wells were set for each sample. The positive controls were the bacterial solution without added antibiotic, and the negative controls were the sterilized MH broth. The absorbance value was measured at OD 600 nm after the 96-well plate was incubated for 24 h at 37 °C. The concentration of antibiotic at which bacterial growth was inhibited entirely was determined to be the MIC [28]. *Escherichia coli* ATCC 25922 was selected as the quality control strain. *Salmonella* Typhimurium was determined to be antibiotic-susceptible, -intermediate or -resistant according to the CLSI M100-S33 standard [29].

### 2.5. RNA Extraction and Library Sequencing

The strain obtained in Section 2.2 was washed twice with PBS (Beijing Solarbio Science & Technology, Beijing, China) buffer, and approximately 10^7^ CFU bacterial precipitates was taken to extract total RNA for subsequent experiments, using the SteadyPure Universal RNA Extraction Kit (AG, Accurate Biotechnology (Hunan) Co., Ltd., Changsha, China).

Transcriptome sequencing was entrusted to Novogene Corporation (Tianjin, China). First, RNA degradation and contamination were monitored on 1% agarose gels (Thermo Fisher Scientific Inc., Waltham, MA, USA), and the integrity was assessed using the RNA Nano 6000 Assay Kit of the Bioanalyzer 2100 system (Agilent Technologies, Santa Clara, CA, USA). And then, mRNA was purified from total RNA using probes (AG, Accurate Biotechnology (Hunan) Co., Ltd., Changsha, China) to remove rRNA. Subsequently, mRNA was randomly interrupted into fragmented templates under Fragmentation Buffer (AG, Accurate Biotechnology (Hunan) Co., Ltd., Changsha, China), and cDNA was synthesized under the M-MuLV reverse transcriptase system (AG, Accurate Biotechnology (Hunan) Co., Ltd., Changsha, China). AMPure XP beads (AG, Accurate Biotechnology (Hunan) Co., Ltd., Changsha, China) were used to screen cDNAs of 370~420 bp that were selected; PCR amplification was performed, and the PCR products were purified again using AMPure XP beads to obtain the library finally.

At last, PCR products were purified (AMPure XP system), and library quality was assessed using the Agilent Bioanalyzer 2100 system. The clustering of the index-coded samples was performed on a cBot Cluster Generation System using TruSeq PE Cluster Kit v3-cBot-HS (Illumia, San Diego, CA, USA) according to the manufacturer’s instructions. After cluster generation, the library preparations were sequenced on an Illumina Novaseq platform, and 150 bp paired-end reads were generated.

### 2.6. Transcriptome Analysis

The clean reads used for the analysis were obtained after raw data filtering, sequencing error rate checking and GC content distribution checking. The clean reads were mapped to the reference genome downloaded from the NCBI Reference Sequence: NZ_CP102669.1. HTSeq v0.6.1 was used to correct the initial read value to FPKM (expected number of Fragments Per Kilobase of transcript sequence per Millions base pairs sequenced) to indicate gene expression. A differential expression analysis of two groups (WT5.4 group vs. ∆*phoP5*.4 group) was performed using the DESeq R package (1.18.0). The resulting *p*-values were adjusted using Benjamini and Hochberg’s approach to control the false discovery rate. Genes with adjusted *p*-value < 0.05 and |log2 (fold change)| > 1 found by DESeq were assigned as differentially expressed genes (DEGs). A Gene Ontology (GO) enrichment analysis of differentially expressed genes was implemented by the GOseq R package, in which gene length bias was corrected. GO terms with corrected *p*-values less than 0.05 were considered significantly enriched by differentially expressed genes. KOBAS 3.0 software was used to test the statistical enrichment of differential expression genes in KEGG pathways.

### 2.7. Determination of Relevant Differential Gene Expression

The extraction of total RNA was performed by referring to Section 2.5. The total RNA was reverse-transcribed into cDNA according to the condition of the *Evo M-MLV* RT Mix Kit with gDNA Clean for qPCR Ver.2 (AG, Accurate Biotechnology (Hunan) Co., Ltd., Changsha, China). Then, the relative expression of the genes was measured by the configured RT-qPCR system of the SYBR^®^ Green Premix *Pro Taq* HS qPCR Kit (AG, Accurate Biotechnology (Hunan) Co., Ltd., Changsha, China). Based on the transcriptome results, 9 DEGs were selected for validation. The sequence information of the genes and primers is listed in Table 1. 16S rRNA was used as a reference gene to normalize the gene expression, RT-qPCR was performed with CFX 96 (Bio-Rad, Hercules, CA, USA) and the Ct values obtained used the 2^−∆∆Ct^ method to obtain the relative gene expression [30]. 

### 2.8. Determination of Negative Cell Membrane Surface Charge

The determination of the negative cell membrane surface charge was performed using an experimental characterization of cell binding to positively charged cytochrome c (Beijing Solarbio Science & Technology, Beijing, China) [32]. A total of 1.8 mL of the strain obtained from Section 2.2 was washed twice with 1 × 3-Morpholinopropanesulfonic acid (MOPS, Beijing Solarbio Science & Technology, Beijing, China) buffer to remove the medium components. The bacterial precipitate collected by centrifugation was resuspended in half the volume of cytochrome c solution (0.5 mg/mL cytochrome c dissolved in 1× MOPS buffer). The obtained bacterial solution was incubated in dark conditions for 10 min, centrifuged at 12,000 rpm for 3 min and 200 µL of the supernatant was aspirated into a 96-well plate. The absorbance value was measured at OD 530 nm with a microplate reader. MOPS buffer without added cytochrome c was used as 100% adsorbed, and 0.5 mg/mL of cytochrome c without added bacteria was used as 100% unabsorbed. The negative charge on the surface of the cell membrane was indicated according to the binding of bacteria and cytochrome c.

### 2.9. The Persistence of Drug Resistance in the Adapted Salmonella Strains during Low-Temperature Storage in ME

Contaminated *Salmonella* may undergo a long and low-temperature maturation process together with the beef. In order to investigate the growth and decline of CAMP resistance in acid-adapted *Salmonella* during storage, meat extract (ME) was utilized as a culture medium to mimic the nutritional and environmental conditions experienced by *Salmonella* during beef storage. ME was prepared based on the method described previously [33]. Beef from the sirloin area was trimmed of fat and tendons, cut into uniformly sized pieces (2 × 2 × 2 cm) and boiled for 30 min at 1:2 beef to deionized water. The meat residue was preliminarily filtered with gauze and then extracted with a vacuum pump. The initial pH of the beef broth medium was 6.25, a portion of the broth was adjusted to pH 5.4 with lactic acid (acidic ME) and another was adjusted to pH 7 with phosphate buffer (neutral ME).

The strains obtained in Section 2.2 were inoculated in acid ME and natural ME, respectively, and stored at 4 °C. The changes in viable bacteria in the ME medium were measured at 0, 6, 13, 20 and 27 d of storage; the drug resistance was measured at 1, 7, 14, 21 and 28 d of storage.

### 2.10. Statistical Analysis

All experiments were performed in at least three independent replications, and the results were expressed as the mean ± standard error. The *D*-value for the survival of bacteria in stressful environments was calculated using the general linear model. Differential gene analysis for transcriptomics sequencing was performed using DESeq R package (1.18.0) software, and gene expression was detected using CFX Manager 3.1 software. Data were analyzed using mixed model analysis with SAS 9.0 and graphing software using Origin 2022 and Prism 8. *p* values < 0.05 were considered statistically significant differences.

## 3. Results

### 3.1. The Effect of Acid Adaptation and ∆phoP on the Various Resistances of S. typhimurium

Regarding acid tolerance, the *D*-value of *S. typhimurium* increased from 46.14 to 81.88 min after acid adaptation (Figure 1A, *p* < 0.05). There was also a significant ATR phenomenon for *phoP*-deficient strains, with the *D*-value increasing from 54.87 to 92.7 min after induction (Figure 1A, *p* < 0.05). No statistically significant difference was found in *D*-values between the WT and ∆*phoP* strains in both the non-adapted and acid-adapted groups.

The results showed a significant increase in the heat tolerance of the WT strain after acid adaptation with *D*-values of 3.23 and 5.49 min, respectively. In contrast, the ∆*phoP* strain lost the ability to increase its heat resistance after acid adaptation with *D*-values of 3.64 and 4.08 min, respectively (Figure 1B). The deletion of *phoP* did not affect heat resistance in non-acid conditions when compared to WT strains. However, under acid-adapted conditions, *phoP* deletion led to a significant decrease in heat resistance (*p <* 0.05).

Figure 2 shows the changes in the number of colonies of WT and ∆*phoP* stored in 8% NaCl before and after acid adaptation. Non-adapted WT strain colony counts decreased significantly during 0–4 days of storage, reaching a steady state at 4 days, decreasing from 6.36 log CFU/mL to 4.19 log CFU/mL. The acid-adapted WT strain colonies showed the least decline, with a final colony count of 4.6 log CFU/mL.

As shown in Table 2, the MIC of WT to polymyxin B without acid adaptation treatment was 4 μg/mL, which increased to 32 μg/mL after acid adaptation treatment, raising it to the drug-resistant type. The MIC of polymyxin B to ∆*phoP* was 2 μg/mL, which showed a decrease compared to WT. Acid adaptation did not affect the MIC of polymyxin B with ∆*phoP*. Similarly, acid adaptation treatment increased the MIC of WT to both the cephalosporin drug ceftazidime and the aminoglycan drug gentamicin, but their susceptibility did not change (Table 2). The MIC of the penicillin drug ampicillin to WT was not affected by acid-adapted treatments, whereas the deletion of the *phoP* gene decreased its MIC. WT and ∆*phoP* were susceptible to ceftazidime and ampicillin before and after acid adaptation (Table 2).

### 3.2. Transcriptomic Analysis

#### 3.2.1. The phoP Gene Plays a Widely Regulated Role in Acid Adaptation

The effect of *phoP* gene deletion on whole gene transcription in WT under acid adaptation environments was analyzed. The quality information of sequencing results is shown in Appendix A, demonstrating the accuracy and reliability of sequencing results. After a comparative analysis, the acid-adapted Δ*phoP* strain had 452 DEGs compared to the acid-adapted WT; 178 genes were up-regulated, and 274 genes were down-regulated (Figure 3).

#### 3.2.2. Determining the Function of DEGs by GO and KEGG Pathway Analysis

After comparing the differential genes between wild-type strains and gene knockout strains, the top ten functional annotations were selected in each of the biological processes (BPs), cellular components (CCs) and molecular functions (MFs). A total of 178 DEGs were enriched in BPs, and the top gene-relative terms were cellular secretion, pathogenesis, cellular protein modification processes and the transmembrane transport of amino acids (Figure 4). All the DEGs regarding the cellular protein modification processes were down-regulated (*p* < 0.05). Regarding the CC process, the 80 DEGs enriched were all related to regulating the cell membrane. The genes associated with the external encapsulating structure, cell outer membrane and external encapsulating structure part were significantly down-regulated (*p* < 0.05). Among the 175 DEGs that were enriched in the MF process, genes related to resistance to acidic environments undergo down-regulation, such as amino acid transmembrane transporter activity, binding to cations’ metal ions and redox activity (*p* < 0.05). Genes related to energy metabolism are up-regulated, such as binding to carbohydrates (*p* < 0.05).

Compared to WT strains, DEGs obtained in the ∆*phoP* strains were submitted to KEGG enrichment analysis, and the top 20 enriched pathways were identified. After deleting the *phoP* gene, the down-regulated DEGs were mainly enriched in *Salmonella* infection, CAMP resistance and quorum sensing (*p*_adj_ < 0.05) (Figure 5). The up-regulated KEGG pathways were concentrated in the bacterial tricarboxylic acid cycle, including starch and sucrose metabolism, lysine degradation, glycolysis/gluconeogenesis, pyruvate metabolism and the pentose phosphate pathway.

The classification of genes according to DEG function showed that *phoP* deletion resulted in the down-regulation of cell membrane composition (regulation of drug resistance) and two-component system gene regulation-related pathways. However, genes in biofilm formation, general cellular resistance and cellular metabolism were up-regulated (Figure 6). Moreover, we randomly selected some differential genes to correlate their RNA-seq and RT-qPCR expressions; the trend of RT-qPCR results was consistent with the transcriptome, and Pearson’s correlation coefficient R^2^ was 0.85, indicating the accuracy of the RNA-seq results (Figure 7).

### 3.3. Antimicrobial Peptide Resistance-Related Gene Expression after Acid Adaptation

To determine the cell membrane modifications by the PhoP/Q system after sensing the acidic environment, the expression of five genes, named *arnA*/*T*, *eptA*, *pagP* and *slyB*, was examined by RT-qPCR. The *arnA/T* and *eptA* genes are responsible for the addition of 4amino-4deoxy-L-arabinose (L-4Arn) and phosphoethanolamine (pEtn) modifications to the cell membrane surface, respectively. After acid adaptation, the expression levels of these three genes in WT were up-regulated by 3.4-, 3.9- and 4.0-fold changes, respectively. After the *phoP* gene was deleted, gene expression in the acid-adapted strain was significantly down-regulated (*p <* 0.05, Figure 8). For the non-adapted strains, the knocking out of the *phop* gene also led to a down-regulation of the three genes. Adding positively charged groups to lipid A reduces the net negative membrane surface charge and hinders cationic drug binding. The genes *pagP* and *slyB* regulate cell membrane fluidity and integrity, respectively. Acid adaptation did not affect the expression of these two genes, and the deletion of the *phoP* gene resulted in a significant down-regulation of gene expression (Figure 8).

Changes in the expression of antibiotic degradation-related genes *pgtE* and *degP* are shown in Figure 9. The expression of the gene *pgtE* was significantly down-regulated after acid adaptation in the WT strain (*p <* 0.05), whereas no significant change was observed in the expression of *degP* (*p* > 0.05). The expression of *pgtE* and *degP* was significantly down-regulated after *phoP* gene deletion (*p <* 0.05). In ∆*phoP*, the expression of *pgtE* and *degP* did not change significantly before and after acid adaptation (*p* > 0.05). There was a 1.7 and 1.6 times up-regulation of *baeR* and *tolC* in the acid-adapted WT strain (*p <* 0.05). In ∆*phoP*, the expression of both genes was not affected by acid adaptation treatments, and the expression of *baeR* was significantly less than in WT (*p <* 0.05).

### 3.4. Acid Adaptation and Gene phoP Significantly Affect Salmonella Cell Membrane Surface Charge

Binding experiments were performed using positively charged cytochrome and WT/∆*phoP* to verify the changes in the surface charge of cell membranes. As shown in Table 3, the acid-adapted treatment significantly increased the unbound rate of WT/∆*phoP* and cytochrome c (*p <* 0.05). After *phoP* gene deletion, the unbound rate of cytochrome c significantly decreased (*p <* 0.05).

### 3.5. Effect of Acid Adaptation and Gene phoP on Salmonella Drug Resistance during Low-Temperature Storage

Meat extract (ME) was utilized as a culture medium to better simulate the nutritional and environmental conditions experienced by *Salmonella* during beef storage [25]. There was no significant change in the number of WT and ∆*phoP* colonies during low-temperature storage, which showed a slow decreasing trend (Figure 10).

The changes in CAMP resistance during storage are shown in Table 4. In neutral ME, the MIC of polymyxin B to the acid-adapted WT strain rapidly decreased to 4 μg/mL on day 1 (After acid adaptation, the MIC of WT to polymyxin B was 32 μg/mL, Table 2), and there was a decreasing trend during subsequent storage. In contrast, the MIC of polymyxin B to WT in acidic ME showed a higher level and a less pronounced decline. In ∆*phoP*, the basal resistance was low (Table 2), remained low and decreased slowly during storage in acidic and neutral ME. Non-adapted WT increased CAMP resistance during acid ME storage but less than pre-acid-adapted WT. Resistance to WT in acidic ME would persist for at least 21 days, with a marked reduction occurring at 28 days, being in a state of resistance. In contrast, ∆*phoP* does not develop resistance in acidic ME, always being in an intermediate state.

## 4. Discussion

Microorganisms have shown a remarkable ability of adaptation in various biotic (e.g., animal skin, foods, etc.) or abiotic (e.g., food contact surfaces) environmental conditions. In fact, the survival as well as the dominance of particular organism are the result of its ability of being able to have implicit properties or develop specific strategies, which allow it to acquire numeric superiority in the niches that develop from the interplay of the physicochemical properties of the biotic and/or abiotic storage conditions in space and time [19].

Controlling their growth is of great importance particularly in the meat industry. Indeed, the reduction in pathogens could minimize the burden of foodborne disease and indirectly affect economic losses. Thus, the interest in improving the microbiological status of meat is vital. Among the decontamination treatments, either physical or chemical treatments or even their combination is commonly used in many countries. The chemical treatment is related to the application of weak acids. The latter creates an acidic environment which is the most commonly encountered stressful environment for *Salmonella* contamination in beef cattle slaughter abattoirs [34]. Bacteria have evolved various effective mechanisms for sensing potential risks and responding to lethal environments, such as acids, antibiotics, high temperatures and organic solvents [35]. This study showed that stationary phase WT developed acid tolerance to extremely acidic environments after a short period of acid adaptation in moderate lactic acid environments (Figure 1A). This phenomenon, known as ATR, was previously found in *Salmonella*, *E. coli* and *Listeria monocytogenes* [36,37]. Alvarez-Ordonez et al. [38] found that *Salmonella* produces different levels of ATR at different organic acids and adaptation temperatures.

Furthermore, the current study also found that acid adaptation increased the ability of WT to survive high temperatures of 55 °C and 8% hypertonic environments (Figure 1B and Figure 2). Ye et al. [8] found that acid adaptation increases heat and osmosis tolerance in *Salmonella enteritidis*. In previous studies, acid adaptation on osmotic tolerance has been inconclusive. Greenacre and Brocklehurst [39] compared the resistance of *Salmonella* to NaCl and KCl after acetic and lactic acid induction and showed that acetic acid induction increases resistance to salt, whereas lactic acid induction causes *Salmonella* to be more sensitive to salt. At the same time, *Salmonella enterica* induced using hydrochloric acid increases salt resistance [8]. In summary, acid adaptation is known to increase the acid resistance of bacteria, but there are differing opinions regarding the impact of such adaptation on multiple stress resistance [40]. A variety of factors, such as strain variability, the acid’s pKa species, acid adaptation time and temperature, may cause this difference [41]. The mechanism of the acid-induced generation of multiple resistance remains to be further explored. Another possible shortcoming of this study is that only a standard strain of Salmonella typhimurium was used to determine the acid resistance and multi-stress resistance of the acid-adapted strain, while strains isolated from the environment may exhibit stronger acid resistance and cross-protection. Therefore, subsequent experiments using environmental isolates to induce acid tolerance and cross-stress resistance may provide more data for actual meat safety. A particularly significant finding is that acid adaptation increases the antibiotic resistance of WT, especially for CAMPs (Table 2). CAMP is a substitute drug used when antibiotics are ineffective. The development of CAMP resistance can result in the emergence of uncontrollable multidrug-resistant bacteria, posing a significant threat to public health security [42]. It has been proposed that *Salmonella* reduces cell membrane permeability and fluidity to indirectly inhibit the invasion of antibiotics [43]. This was confirmed by the expression levels of membrane modification genes in acid-adapted WT/∆*phoP* (Figure 8).

The PhoP/Q two-component system in *Salmonella* is an important regulatory mechanism for detecting and transmitting stress signals, which modulate basic cellular physiological processes, secondary metabolism and stress responses in response to environmental changes [44]. In this study, the *phoP* gene was knocked out to investigate the changes in multiple resistance. The results of this study showed that the *phoP*-deficient strains did not significantly increase heat tolerance after acid adaptation (Figure 1B). This suggests that the PhoP/Q system simultaneously mediates the generation of heat resistance in WT during acid adaptation. Unexpectedly, the acid tolerance of ∆*phoP* was not significantly reduced, and the acid tolerance after acid adaptation was similar to WT. It is assumed that there are other critical pathways outside of phoP/Q in the acid-adapted pathway that generates ATR. The following transcriptomics found that the expression of the genes *adiA* and *elaB* of ∆*phoP*, encoding arginine decarboxylase and stress response proteins, respectively, was significantly up-regulated after acid adaptation. In previous studies, stress protein regulatory gene *nlpD* and sigma factor RpoS also mediated changes in acid tolerance during acid adaptation [45,46].

The current research also suggests that the PhoP/Q system may be a key regulator of CAMP resistance. Enhanced sensitivity to CAMP polymyxin B was observed following the deletion of the *phoP* gene, and the *phoP* deletion strain did not develop resistance when it was adapted to the acidic environment again. As for ceftazidime and Gentamycin, although the resistance may reduce after *phoP* gene deletion, the MIC returned to the same level as the WT strain after acid adaptation. These different resistance patterns reveal that acid adaptation and the *phoP* gene enhance the resistance of different antibiotics through their own pathways and especially the *phoP* gene is more closely associated with CAMP resistance. In summary, the signaling and regulation of WT against acidic environments is complex, with cross-protective effects such as heat resistance, thermotolerance and drug resistance mediated by the PhoP/Q system. The ATR phenomenon caused by acid adaptation to the environment has a large variability, and attention should be paid to the setting of the rest of the fencing factor parameters after non-lethal acid adaptation in the production process.

As PhoP/Q may be associated with a wide range of resistances but its specific regulatory mechanisms were still unclear, transcriptomics was used to analyze the molecular mechanisms by which the PhoP/Q system functions during acid adaptation. In this study, we found that WT modulates CAMP resistance through the phoP/Q system after acid adaptation. Similar to our results, the resistance of CAMPs increased after acid adaptation, but the regulatory mechanism of the increase in CAMP resistance has not been deeply explored [47]. The results of the GO enrichment analysis showed that the most significant gene expression changes were in virulence, which is consistent with the study that the gene *phoP* controlled the formation of virulence islands [48]. Genes involved in amino acid transmembrane transport were identified as down-regulated. It has been demonstrated that bacteria utilize amino acid de-carboxylation to consume internal protons when protons from the external environment enter the cell and disrupt internal pH homeostasis [49]. The PhoP/Q system plays a crucial role in maintaining internal pH homeostasis by regulating amino acid metabolism, which is essential for its role in the acid tolerance response mechanism [25,50].

To investigate the pathways affected by the PhoP/Q system on cell membrane structure and CAMP resistance, DEGs related to cell membrane modification were analyzed. This study confirmed that the WT-increased CAMP resistance during acid adaptation was associated with cell membrane modification pathways. The genes *arnA*, *arnT*, *eptA*, *pagP* and *slyB*, known to be involved in cell membrane modification, were validated using RT-qPCR. There was a 3–4-fold enhancement in the relative expression of the genes *arnA*/*T* and *eptA* in WT after acid adaptation, and consistent with the transcriptomic results, the expression of these three genes was down-regulated after *phoP* knockdown. Modifying the outer membrane lipid A is a major means of resistance to CAMPs for Gram-negative bacteria [51]. With the positive charge and hydrophobic acyl chains carried by polymyxin B, it is highly susceptible to electrostatic interaction with bacterial lipopolysaccharides (LPS), displacing divalent cations in the molecule and causing the disruption of the membrane, leading to the leakage of the contents and killing of the bacteria. *Salmonella arnA*/*T* genes encode 4amino-4-deoxy-L-arabinose (L-4Arn) transferase that catalyzes the incorporation of L-4Arn into lipid A, reduces surface charge and impedes the binding of CAMP drugs [52]. Similarly, *Salmonella eptA* encodes the enzyme EptA, which catalyzes a modification of phosphoethanolamine (pEtn) to lipid A 1′position, replacing the original phosphate group and leading to a reduction in net negative charge [53]. Yin et al. [54] found a modification of the lipid A structure by L-4Arn and pEtn in *Aeromonas*, and such modification reduces the net charge on the cell membrane surface significantly. In this study, further measurements of the net negative charge on the cell membrane surface showed a decrease in net negative charge after acid adaptation and an increase in net negative charge after *phoP* knockdown (Table 3). The change in net negative charge on the bacterial surface corresponds to the phenotype of enhanced AMP resistance after acid-adapted treatment and decreased AMP resistance after *phoP* knockdown (Table 2). In addition, it has been found that the expression of the genes *arnA*/*T* and *eptA* is regulated by the two-component system PmrA/B [55]. Moreover, *phoP* indirectly activates the response regulator PmrA of PmrA/B through the downstream regulatory protein PmrD [56]. Accordingly, a pathway process is hypothesized from PhoP/Q sensing the acidic environment to activating PmrA/B, to regulating the expression of genes encoding negatively charged enzymes.

The expression of genes *pagP* and *slyB* was significantly down-regulated after *phoP* knockdown. The gene *pagP* acts to modify lipid A, acting on the outer membrane to acetylate ethyl palmitate to produce ethyl palmitate, which is added to the lipid A 2′position [55]. The purpose of this modification is to enhance the hydrophobic interactions of the outer membrane and reduce mobility, thereby impeding the penetration of CAMPs into the cytosol and the formation of pores. [57]. The gene *slyB* encodes a conserved outer membrane lipoprotein, SlyB, contributing to cell envelope integrity [58]. Notably, after *phoP* deletion, most genes associated with cell membrane composition and modification were down-regulated in the transcriptome results. The transcriptomics results show that the functions of the external encapsulating structure, cell outer membrane and integral component of the membrane are annotated to the *pagP* gene. It was shown that PhoP/Q in *Salmonella* responds to the immune system within the host by activating *pagP* expression and regulating bacterial outer membrane remodeling and outer membrane vesicle production [59]. As the bridge and the first barrier between the cell and the external environment, the cell membrane will be adjusted to change its fluidity according to the external environment to survive in extreme conditions [60].

The efflux and degradation of already bound CAMPs are also critical protective tools for bacteria. In Gram-negative bacteria, multidrug efflux pumps, especially those belonging to the resistance-nodulation-division family, play an important role in drug resistance [61]. AcrAB-TolC is a well-known multidrug efflux pump in bacteria, where AcrB acts as the substrate-binding domain located in the inner membrane, TolC serves as the channel for transporting substrates and AcrA functions as the bridge between the inner and outer membranes [62]. Pérez et al. [61] discovered antibiotic resistance in clinical isolates of *Enterobacter cloacae* and identified the involvement of the drug efflux pump AcrAB-TolC in the mechanism of multiple antibiotic resistance. They also observed that the AcrAB-TolC system was widespread within the species. All of the *Salmonella* isolated from livestock and poultry animal production had drug efflux pump activity, with 68.8% of the strains carrying the AcrAB-TolC system [63]. The down-regulation of the efflux pump gene *tolC* after phoP gene deletion suggests that the PhoP/Q system improves CAMP resistance by increasing the transport capacity of efflux pumps. In previous studies, DegP functions as a protease that recognizes whether a protein is correctly synthesized, thus determining its transport to degradation, repair or entry into the target cell [64]. Ulvatne et al. [65] found that the protease DegP increases resistance to the CAMP lactoferrin B in *Escherichia coli* and *Staphylococcus aureus*. In this study, we found that *phoP* knockdown down-regulated *degP* expression, while acid adaptation had no significant effect on *degP* expression (Figure 9). DegP may act as a protease that is indirectly regulated by PhoP/Q in order to recognize and degrade CAMP molecules as they enter the periplasm. The impact of DegP proteases on CAMP drugs has not been extensively studied, suggesting a potential avenue for future targeted drug design.

The persistence of induced drug resistance in WT during beef storage may pose a significant threat to consumers. In this study, WT was incubated in ME medium (pH = 5.4) at 4 °C to simulate the beef storage environment. Acid-adapted WT maintained CAMP resistance for at least 21 days in ME medium. For non-induced WT, mild resistance was induced during the storage, whereas *Salmonella* after the deletion of the PhoP/Q system was not induced to develop AMP resistance even after prolonged storage in acidic environments. It was also found that adapted WT lost its acquired CAMP resistance after being transferred to neutral conditions (ME medium, pH = 7) in only 1 day. These results indicate that the drug resistance induced by acid tolerance will persist throughout the meat’s storage and sale process causing harm to public health, and more attention should be paid to such risk. However, wild-type *Salmonella* or non-typhoidal *Salmonella* isolated from the environment have different characteristics from the classical reference strains. It is necessary to supplement the acid adaptation variation characteristics of various serotypes of *Salmonella* in subsequent studies to better predict risk assessment.

## 5. Conclusions

Acid adaptation treatments increased the tolerance of *Salmonella* to lethal conditions such as acid, heat, salt and antibiotics. The PhoP/Q TCS was found to be activated by the acidic environment and involved in the regulation of co-occurring heat tolerance and antibiotic resistance. Transcriptomics and RT-qPCR analyses revealed that the deletion of the *phoP* gene resulted in the down-regulation of the expression of genes related to lipid A modification and efflux pumps. These changes in the gene expression may result in the change in net negative charge and the mobility of the cell membrane resulting in an enhanced CAMP resistance. The fact that drug resistance can persist in stimulated beef storage environments reveals the potential threat of the acid tolerance response in foodborne pathogens. These results shed light on the protective mechanisms of the acid-induced environment leading to reduced antibiotic resistance susceptibility in *Salmonella* at the molecular and epigenetic levels. Although the present study confirmed part of the enhanced cross-protection under acid-inducing conditions, the specific regulatory mechanisms involved in the development of multi-stress resistance regulated by the PhoP/Q system remain to be further investigated.

## Figures and Tables

**Figure 1 foods-13-01533-f001:**
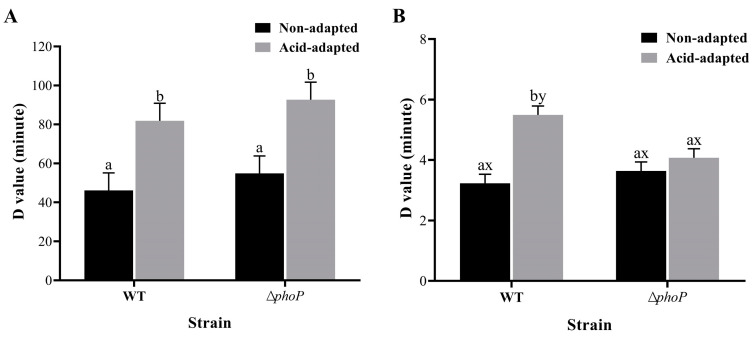
The PhoP/Q system and acid adaptation regulate *Salmonella* survival *D*-values in response to environmental stresses. The *D*-values of acid (**A**) and heat (**B**) stress. ^a,b^ indicate significant differences in different induction treatments (non-adapted and acid-adapted) at *p <* 0.05. ^x,y^ indicate significant differences in different strains (WT and ∆*phoP*) at *p <* 0.05.

**Figure 2 foods-13-01533-f002:**
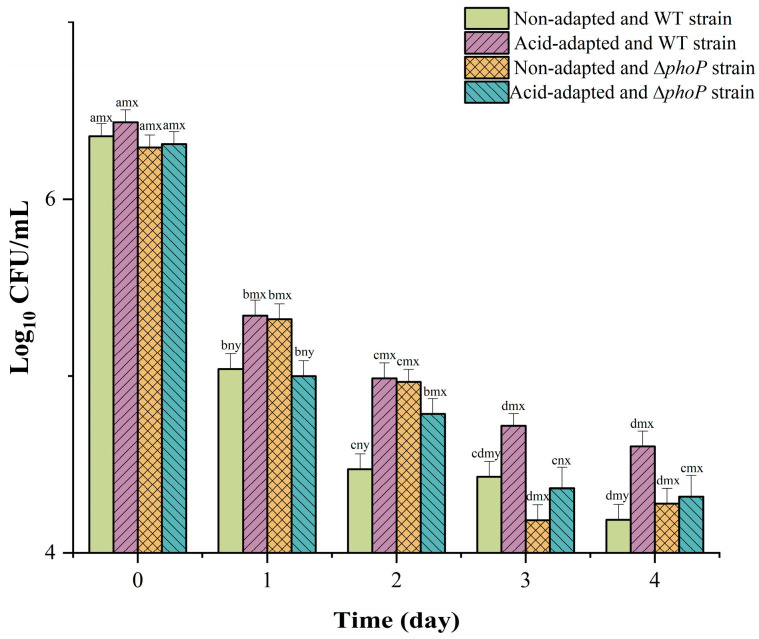
The PhoP/Q system and acid adaptation regulate the logarithmic reduction in *Salmonella* storage at 8% NaCl. ^a–d^ indicate significant differences in different storage time at *p <* 0.05. ^m,n^ indicate significant differences in different strains (WT and ∆*phoP*) at *p <* 0.05. ^x,y^ indicate significant differences in different induction treatments (non-adapted and acid-adapted) at *p <* 0.05.

**Figure 3 foods-13-01533-f003:**
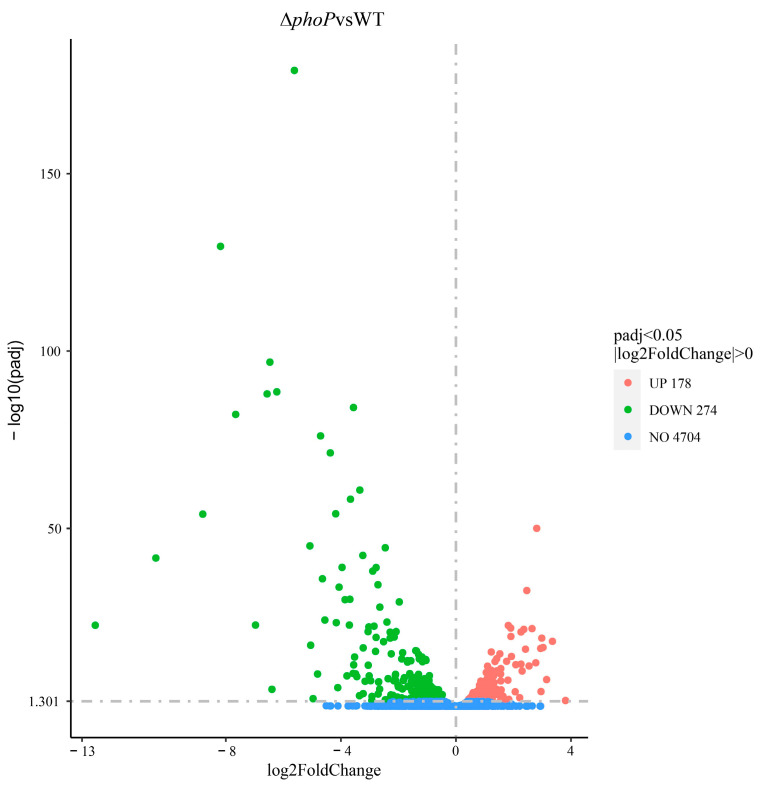
Volcano plot representation of DEGs (acid-adapted ∆*phoP* compared to WT).

**Figure 4 foods-13-01533-f004:**
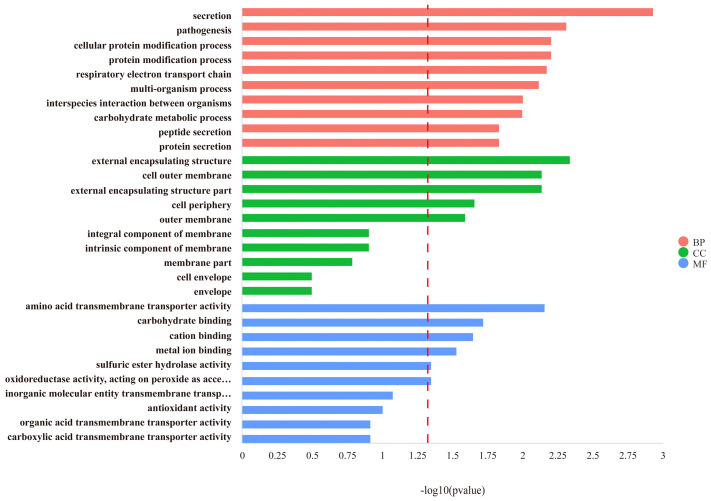
GO enrichment analysis of DEGs (acid-adapted ∆*phoP* compared to WT). The right side of the red dashed line represents the enriched gene function with *p* < 0.05.

**Figure 5 foods-13-01533-f005:**
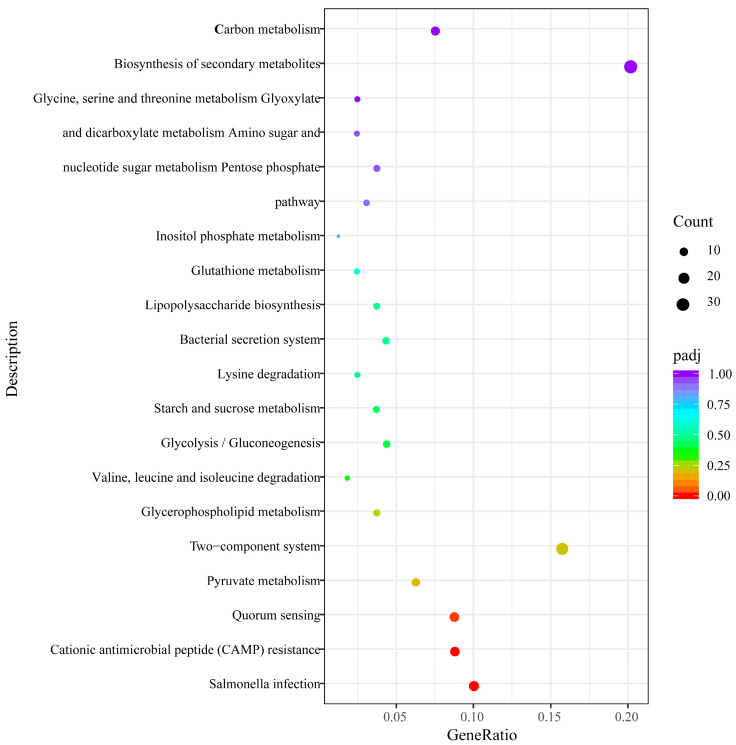
A scatter plot of the KEGG enrichment analysis (∆*phoP* compared to WT after acid adaptation).

**Figure 6 foods-13-01533-f006:**
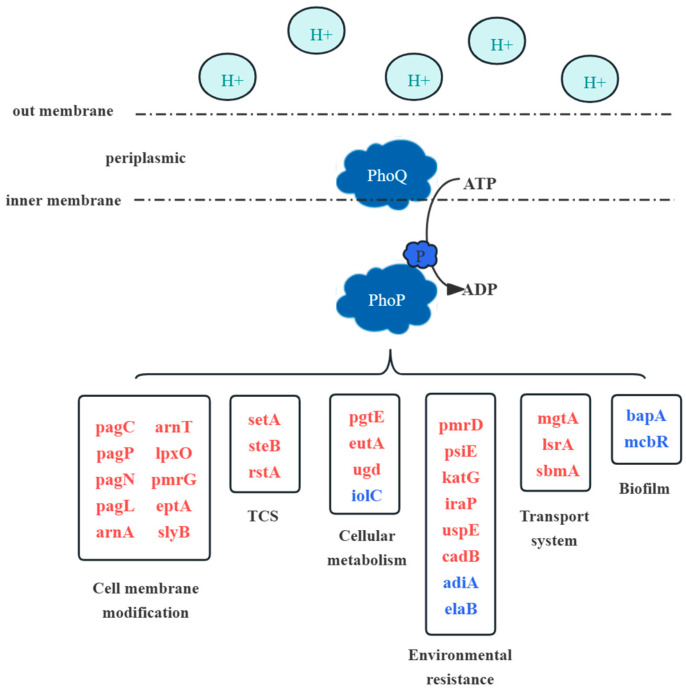
The regulation of downstream genes by the PhoP/Q system in acidic environments. The red font represents down-regulated genes, and the blue font represents up-regulated genes.

**Figure 7 foods-13-01533-f007:**
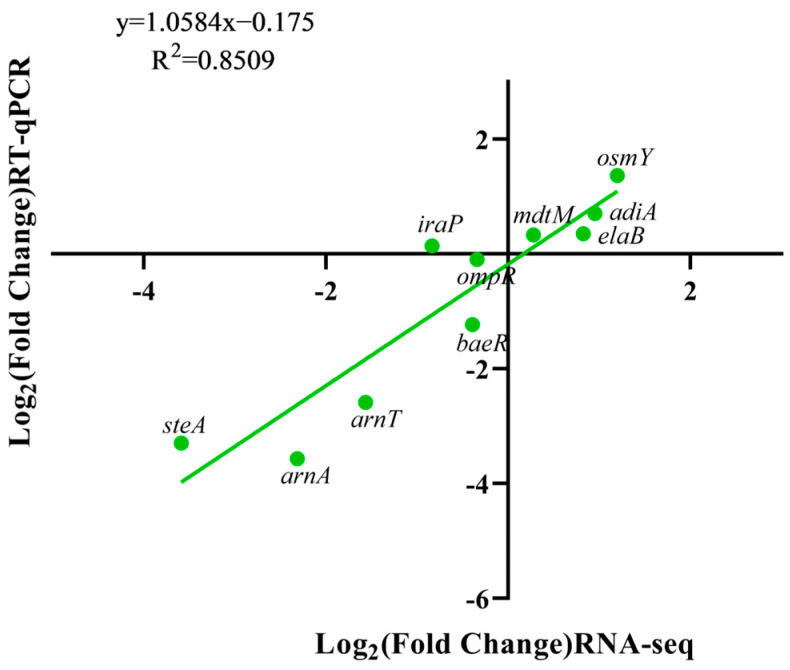
Correlation analysis of WT and ∆*phoP* RT-qPCR with RNA-seq after acid adaptation.

**Figure 8 foods-13-01533-f008:**
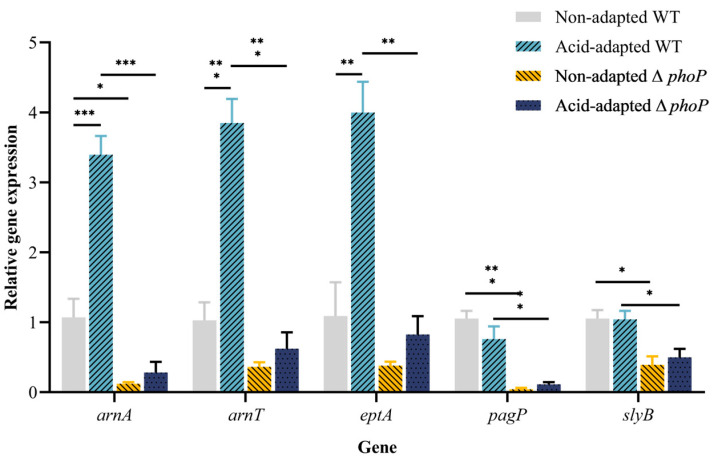
Relative expression of genes *arnA*, *arnT*, *eptA*, *pagP* and *slyB* before and after acid adaptation in WT and ∆*phoP* strains. Error bars represent standard deviations. *, *p <* 0.05; **, *p <* 0.01; ***, *p <* 0.001.

**Figure 9 foods-13-01533-f009:**
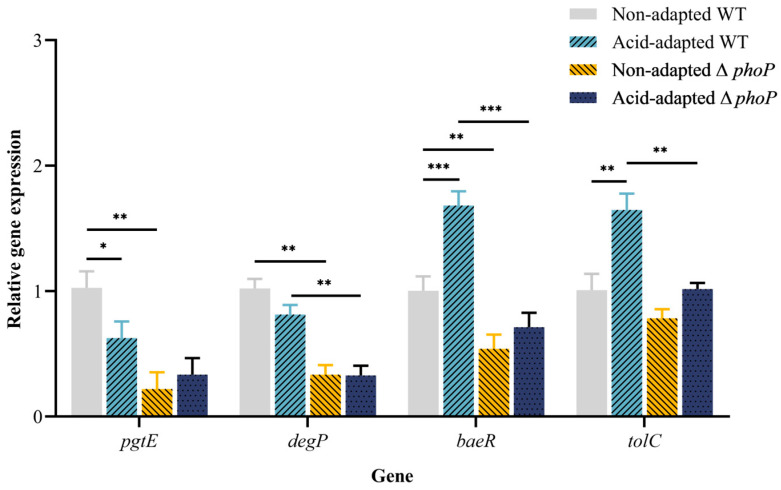
Relative expression of genes *pgtE*, *degP*, *beaR* and *tolC* before and after acid adaptation in WT and ∆*phoP* strains. Error bars represent standard deviations. *, *p <* 0.05; **, *p <* 0.01; ***, *p <* 0.001.

**Figure 10 foods-13-01533-f010:**
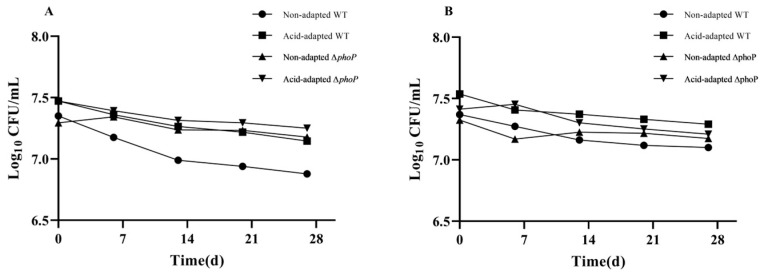
Changes in the number of viable bacteria of neutral ME (**A**) and acidic ME (**B**) media during low-temperature storage (log CFU/mL).

**Table 1 foods-13-01533-t001:** Genes and primer sequences for the real-time PCR.

Gene	Primer Sequence (5′-3′)	Function	References
16S rRNA	F: CAGCCACACTGGAACTGAGA	16 S ribosomal RNA	[31]
	R: GTGCTTCTTCTGCGGGTAAC		
*arnA*	F: ACCAGAACCCAGTTTAGCGG	Bifunctional UDP-4-amino-4-deoxy-L-arabinose formyltransferase	This study
	R: CCGCAACCTGTTAAGCGAAG		
*arnT*	F: TCGGTGCGAAACAGGAAAGA	Lipid IV(A) 4-amino-4-deoxy-L-arabinosyltransferase	
	R: GCGAAACGGCGCATTCTATT		
*eptA*	F: GGCGAGTGCTACGATGAAGT	Phosphoethanolamine transferase EptA	
	R: TGTAATAGGTTGGGCCGTGG		
*pagP*	F: TTCAGTCTCTGCGGCGGATAA	Lipid IV(A) palmitoyltransferase PagP	
	R:GGTAATGGCGGGGACATACAAATC		
*slyB*	F: TTATCCCTAGCGGGGTGTGT	Outer membrane lipoprotein SlyB	
	R:CCCTGAATTTGAACCGGACG		
*pgtE*	F: CGGACACCAGCGTCAACTAT	Omptin family outer membrane protease PgtE	
	R: CGCCTTGTAGTTATCGCCCT		
*degP*	F: TTCACCTGGCCGTATTCCAC	Serine endoprotease DegP	
	R: CTGGTGAACCTGAACGGTGA		
*baeR*	F: TCATTAACGGGCTTTCGGCA	Response regulator in two-component regulatory system with BaeS	
	R: AATCGAAGAGATCGACCGGC		
*tolC*	F: GATACAGCGGCAGGGAGAAG	TolC family outer membrane protein	
	R: CCAACTCCACCCAGTACGAC		

F: forward; R: reverse.

**Table 2 foods-13-01533-t002:** Effect of acid induction and gene *phoP* on MIC (μg/mL) of antibiotics.

Antibiotics	Acid Adaptation Treatment	WT	∆*phoP*
Polymyxin B	Non-adapted	4(I)	2(I)
	Acid-adapted	32(R)	2(I)
Ceftazidime	Non-adapted	0.5(S)	0.25(S)
	Acid-adapted	1(S)	1(S)
Gentamycin	Non-adapted	8(R)	8(R)
	Acid-adapted	16(R)	16(R)
Ampicillin	Non-adapted	2(S)	1(S)
	Acid-adapted	2(S)	1(S)

**Table 3 foods-13-01533-t003:** The unbound rate of cytochrome c assessed the effect of different induction treatments and compositional systems on the surface charge of the cell membrane.

Strain	pH	
	7	5.4
WT	75.95 ± 3.66 ^bx^	81.51 ± 2.65 ^ax^
∆*phoP*	70.15 ± 2.92 ^by^	72.04 ± 2.98 ^ay^

Different letters indicate significant differences between treatment groups after ANOVA (*p <* 0.05). Unbound rate expressed as percentage. ^a,b^ indicate significant differences in different induction treatments (non-adapted and acid-adapted) at *p <* 0.05. ^x,y^ indicate significant differences in different strains (WT and ∆*phoP*) at *p <* 0.05.

**Table 4 foods-13-01533-t004:** Changes in MIC of *Salmonella* during low-temperature storage in ME broth medium (μg/mL).

Storage Time (d)	Medium	Strain and pH			
		WT		∆*phoP*	
		7	5.4	7	5.4
1	Neutral ME	4(R)	4(R)	2(I)	2(I)
	Acidic ME	16(R)	32(R)	1(I)	2(I)
7	Neutral ME	2(I)	2(I)	1(I)	1(I)
	Acidic ME	16(R)	32(R)	1(I)	1(I)
14	Neutral ME	1(I)	2(I)	2(I)	2(I)
	Acidic ME	16(R)	32(R)	1(I)	0.5(I)
21	Neutral ME	2(I)	2(I)	2(I)	2(I)
	Acidic ME	16(R)	16(R)	1(I)	1(I)
28	Neutral ME	1(I)	1(I)	0.5(I)	0.5(I)
	Acidic ME	4(R)	8(R)	0.5(I)	0.5(I)

## Data Availability

The original contributions presented in the study are included in the article/Appendix A, further inquiries can be directed to the corresponding author.

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
