# Peer review of "The Effect of the PhoP/PhoQ System on the Regulation of Multi-Stress Adaptation Induced by Acid Stress in Salmonella Typhimurium"

_foods, 2024, doi:10.3390/foods13101533_

Round 1

Reviewer 1 Report

Comments and Suggestions for Authors

The authors investigated in two Salmonella strains – wild type and deleted phoP mutant - the effect of acid adaptation and the effect of the PhoP/Q system on the regulation of stress environmental tolerance, such as acid, heat, salt, and antibiotic. They analyzed the survivability of the two strains before and after acid adaptation. Additionally, they performed the transcriptome and RT-qPCR analyses. The study was well-designed and carefully conducted. However, there are some issues and I have a few comments for the authors to address:

The enrichment analysis should focus on terms with significant p value or adjusted p value. I recommend reviewing this analysis as well as Figures 4 and 5.

The paragraph on line 301 is not clear. I understand that the analysis was performed using the DEGs obtained in the comparison between WT and the DphoP strain. So, please describe it better.

Describe the TCS acronym.

Figure 4: In the x-axis, the GO terms are eligible, the letters are too small.

Line 329: I understand that the sentence refers to the comparison of acid-adapted WT and non-adapted WT samples, so the fold-change values ​​should be mentioned. Therefore, I recommend separating the results from Figure 8 and better describing the results of the comparisons.

Line 256: “the population changes in the Salmonella”. Be careful with this statement, as it can cause a misinterpretation, as the population did not change, what changed was the bacterial proliferation (CFU).

Be careful when using Salmonella on the Result and Discussion sections (e.g. lines 256, 257, 355, 375, etc.), because you used two Salmonella strains (ATCC14028 and ATCC14028 DphoP mutant), so be specific.

Table 3: n the first column it would be more appropriate to change it to “Strain”. In the second column you could include “(pH)”. Describing the statistical analysis, the comparison is not clear. If the comparison was between two groups, why did you use a multivariate test?

Table 4: In the Component Systems column you could include “(pH)”.

There is a recurring error in the text. When describing the results obtained or the comparisons made, it is necessary to make it clear on which samples the data were obtained. For example, in paragraph 287, in lines 329, 341, 375, etc. Please be careful as it makes results interpretation very difficult.

Line 357: “Deleting the phoP…” could be rewrite to “In the DphoP strain…”

Lines 397 – 401, 485-487, 424-425, 475, 517: please cite some reference.

The Discussion section is very exhaustive, e.g. there is no need to repeat the results. I think the authors should focus on discussing the study findings and could also summarize the description of the cited works.

Rearrange the sentences of the lines 461-467, e.g. "In this study" (line 467) it would be better at the beginning of the sentence of the line 461. ​​Another example is in line 470, “most significant changes” would be more appropriate “most significant gene expression changes were in genes...”

Line 61 and 84: Is the acronym for cationic antimicrobial peptides wrong?

Lines 245 and 247: better to use “strain” than “cell”.

Line 447: “phoP” in italic.

Author Response

The authors investigated in two Salmonella strains – wild type and deleted phoP mutant - the effect of acid adaptation and the effect of the PhoP/Q system on the regulation of stress environmental tolerance, such as acid, heat, salt, and antibiotic. They analyzed the survivability of the two strains before and after acid adaptation. Additionally, they performed the transcriptome and RT-qPCR analyses. The study was well-designed and carefully conducted. However, there are some issues and I have a few comments for the authors to address.

Response: Thanks for the carefully review and positive feedback on our manuscript. The comments have greatly improved our manuscript. Careful revisions have been made throughout the manuscript, and more detailed interpretation had been added as suggested by the reviewer.

The enrichment analysis should focus on terms with significant p value or adjusted p value. I recommend reviewing this analysis as well as Figures 4 and 5.

Response: Thanks for the comments, we marked the critical P value 0.05 with dotted lines in Figure 4 and the -log10(p) can be found in the X-axis. The adjust P value can be found in Figure 5. In addition, we have modified the descriptions on revised manuscript line 309, 312, 314-316 and 321-323 to get a clear expression.

The paragraph on line 301 is not clear. I understand that the analysis was performed using the DEGs obtained in the comparison between WT and the DphoP strain. So, please describe it better.

Response: Thanks for the suggestions, sentence had been rewrote to get an accurate expression and on revised manuscript line 303 and 320-321.

Describe the TCS acronym.

Response: We are sorry for the carelessness, the full name of TCS and ATR has been completed, on line 22 and 71.

Figure 4: In the x-axis, the GO terms are eligible, the letters are too small.

Response: Thanks for the suggestions, Figure 4 was redrawn.

Line 329: I understand that the sentence refers to the comparison of acid-adapted WT and non-adapted WT samples, so the fold-change values ​​should be mentioned. Therefore, I recommend separating the results from Figure 8 and better describing the results of the comparisons.

Response: Thanks for the comments, we describe the functions and fold differences of the genes in Figures 8 separately to present the results more clearly and on revised manuscript line 348-352.

Line 256: “the population changes in the Salmonella”. Be careful with this statement, as it can cause a misinterpretation, as the population did not change, what changed was the bacterial proliferation (CFU).

Response: Thanks for the suggestions, we have replaced the misunderstood statements and on revised manuscript line 270-274.

Be careful when using Salmonella on the Result and Discussion sections (e.g. lines 256, 257, 355, 375, etc.), because you used two Salmonella strains (ATCC14028 and ATCC14028 DphoP mutant), so be specific.

Response: We are sorry for the carelessness, we replaced the ambiguous expression of Salmonella in the Results and Discussion with WT and ∆phoP to clearly distinguish the results and on revised manuscript line 270, 271, 376, 377, 428, 434, 447, 452, 460, 477, 486, 500, 568 and 573.

Table 3: n the first column it would be more appropriate to change it to “Strain”. In the second column you could include “(pH)”. Describing the statistical analysis, the comparison is not clear. If the comparison was between two groups, why did you use a multivariate test?

Response: Thanks for the suggestions, we modified the first row of Table 3. In addition, we modified the data analysis method of Table 3. Some data analysis results were modified in lines 378-379 of the revised manuscript.

Table 4: In the Component Systems column you could include “(pH)”.

Response: Thanks for the suggestions, we modified the first row of Table 4.

There is a recurring error in the text. When describing the results obtained or the comparisons made, it is necessary to make it clear on which samples the data were obtained. For example, in paragraph 287, in lines 329, 341, 375, etc. Please be careful as it makes results interpretation very difficult.

Response: Thanks for the suggestions, we have checked and revised the similar description and on the revised manuscript line 303-304, 348-352, 355-356, 363-365, 397-398, 452-453, 534 and 569.

Line 357: “Deleting the phoP…” could be rewrite to “In the DphoP strain…”

Response: Thanks for the suggestions, modification had been made as suggested and on the revised manuscript line 378-379.

Lines 397 – 401, 485-487, 424-425, 475, 517: please cite some reference.

Response: Thanks for the suggestions,we have supplemented the references according to the suggestions, and the reference numbers in the revised manuscript were 33, 31 ,48 and 50. The results of line 517 were derived from transcriptome sequencing results, and no references were added in the revised manuscript.

The Discussion section is very exhaustive, e.g. there is no need to repeat the results. I think the authors should focus on discussing the study findings and could also summarize the description of the cited works.

Response: Thanks for the comment. We reduced unnecessary results in the discussion section according to the suggestions.

Rearrange the sentences of the lines 461-467, e.g. "In this study" (line 467) it would be better at the beginning of the sentence of the line 461. ​​Another example is in line 470, “most significant changes” would be more appropriate “most significant gene expression changes were in genes...”

Response: Thanks for the suggestions, we have adjusted the logic of the description according to the suggestions and on the revised manuscript line 485-487.

Line 61 and 84: Is the acronym for cationic antimicrobial peptides wrong?

Response: We are sorry for the carelessness, the acronym for cationic antimicrobial peptides has been corrected and on the revised manuscript line 59, 82 and 532.

Lines 245 and 247: better to use “strain” than “cell”.

Response: Thanks for the suggestions, we have changed the wording to be more accurate based on your suggestions and on revised manuscript line 259 and 261.

Line 447: “phoP” in italic.

Response: We are sorry for the carelessness, we corrected the formatting of this gene name and on revised manuscript line 471.

Reviewer 2 Report

Comments and Suggestions for Authors

Authors choose good model bacteria, Salmonella Typhimurium strain CDC 6516-60 (ATCC, 14028i) isolated from the pooled heart and liver tissue of four-week-old chickens. This bacterium has applications in enteric disease research, media testing, and water testing.  

Major comments:

The novelty of this study is not clear; please elucidate the novelty in the light of the gap between your study and previous reports Authors described almost the same study in previous article:

Yunge Liu , Yimin Zhang , Lixian Zhu, Lebao Niu , Xin Luo , Pengcheng Dong

The acid tolerance responses of the Salmonella strains isolated from beef processing plants. Food Microbiol. 2022 Jun;104:103977. doi: 10.1016/j.fm.2022.103977. Epub 2022 Jan 7. PMID: 35287806.

In above article Salmonella strains isolated from beef processing plants (different serotypes, S. Kottbus, S. Derby, S. Meleagridis, S. Agona, S. Calabar, S. Senftenberg, S. Kingston) were examined. In this study authors examined only 1 reference strain and phoP gene deletion mutation strain (ΔphoP).

Authors previously described and examined S. Kottbus, S. Derby, S. Meleagridis, S. Agona, S. Calabar, S. Senftenberg, S. Kingston isolated from beef processing plants which might be added to  this study. Why S. Typhimurium was chosen?

Abstract – a little unclear, spme information about Salmonella, cationic antimicrobial peptide (CAMP), not detailed enough and chaotic. It should be improved.

The Authors are encouraged to add a schematic illustration, presenting the steps conducted in this study to facilitate the following of the current investigations.

Line 105 - Acquisition of acid-adapted strains, what control strains were used?

 Line 111 -… and control strains were obtained by water bath at 37 °C for 90 min for subsequent experiments… - sentence is unclear, please revised

Line 113 - 2.3 Determination of acid, heat and salt stress under different conditions of pH and strains - strain obtained in part 2.2 (WT and ∆phoP) and control strains?

2.3 Determination of acid, heat and salt stress under different conditions of pH and strains

Line 123 - 126 – methodology based on reference 28 – where Determination of heat resistance were prepared in TSB not in LB, and according completely different methods.

2.4 Determination of  antibiotic resistance under different conditions of pH and strains

The MICs were interpreted according to the CLSI,  EUCAST or other comitee? It is impotrant especialy for florfenicol. Plase add details. Lack of polymyxin B, peftazidime, gentamycin, florfenicol, ampicillin concentration. Why were the five antibiotics chosen?

Line 139 - …Less than 5% Dimethyl sulfoxide (DMSO) is added as a co-solvent to antibiotic  solutions that are not readily soluble in water – pleas add this antibiotic names and add  details about study about bacteriostatic effect concentration of DMSO.

How many replicates did you apply for the antibacterial assays? According which guistandards for antimicrobial susceptibility testing MIC methods were performed and why LB broth was used? Most common Medium is Cation-adjusted Mueller-Hinton broth.

In paragraph lack of information about antibiotic resistance under different conditions of pH (and strains – please revised sentence). Moreover, cited reference 29 (to line 145-146 or for whole paragraph) described: Effects of clove essential oil and oregano essential oil at sub-MICs on the biofilm formation of S. Derby, but not MIC for antibiotics.

2.5 RNA extraction and library sequencing

Line 150 – what is mean: and appropriate  amounts of bacterial precipitates

Lack of Quality control strain to MIC methods

2.9 Determination of drug resistance during storage at low temperature – paragraph is completely unclear. Determination of drug resistance in Cooking of Meat extract (ME)?

Results

Line 266 – 268 – concentration of antibiotics should be calryied in methods eg. polymyxin B 4 μg/mL to 32 μg/mL, 4 μg/mL to 2 μg/mL. Lack of origin concentration. Moreover lack of interpretation of MIC results - strains was resistant or susceptible?

Figure 4. description of x – axis is completely unreadable

Line 313 – which drug-resistance related pathways genes were downregulated?

3.5 Effect of acid adaptation and gene phoP on Salmonella drug resistance during low-temperature storage

What is the link between meat extract (ME) used as an environmental substrate at low temperature and drug resistance?

Lack of reagents producer - non-glucose tryptic soy broth (NTSB), glicerol, lactic acid, petone, NaCl, hydrochloric acid, DMSO, PBS, agarose, probes to remove rRNA, Fragmentation Buffer, M-MuLV reverse transcriptase system, AMPure XP beads, 3-Morpholinopropanesulfonic acid, cytochrome c solution

I  have critical points regarding self citation of Xu Gao (1 self citation as first author, 1 as co-author), Jina Han (1 self citation as first author), Lixian Zhu (3 as co-author), George-John E. Nychas (3 as co-author), Yanwei Mao (1 as co-author), Xiaoyin Yang (2 as co-author), Yunge Liu (3 self citation as first author), Yimin Zhang (5 as co-author), Pengcheng Dong (4 as co-author). Authors cited own previus study. It must be changed.

Some of the cited articles are outdated. 38 from 61 older than 5 years.

Minor comments:

Line 19 - Salmonella Typhimurium instead of Salmonella typhimurium

Line – 22 ATR - expand abbreviation

Line 107, 111 - 37°C instead of 37 °C

Line 109  - 4°C

Line 110, 112, 118 – min.

Line 125, 410 -  55°C

Line 144 - added antibiotic insted of, added inhibitor,

Author Response

Authors choose good model bacteria, Salmonella Typhimurium strain CDC 6516-60 (ATCC, 14028i) isolated from the pooled heart and liver tissue of four-week-old chickens. This bacterium has applications in enteric disease research, media testing, and water testing.

Major comments: The novelty of this study is not clear; please elucidate the novelty in the light of the gap between your study and previous reports Authors described almost the same study in previous article:

Yunge Liu, Yimin Zhang , Lixian Zhu, Lebao Niu , Xin Luo , Pengcheng Dong

The acid tolerance responses of the Salmonella strains isolated from beef processing plants. Food Microbiol. 2022 Jun;104:103977. doi: 10.1016/j.fm.2022.103977. Epub 2022 Jan 7. PMID: 35287806.

Response: Thanks for the question. The current research is a further deepening of our previous study conducted by Liu et al. In the previous study, we mainly focus on the effect of different pH (5.0, 5.4, 6.0, and 7.0), temperatures (10 â—¦C and 37 â—¦C), adaptation media (meat extract and brain heart infusion media) and strain differences on the development of acid tolerance response. The possible induction of ATR during beef production was confirmed. On the base of that study, expect the acid tolerance, our current study design to confirm if the “cross protection” i.e. whether the heat, osmotic, and antibiotic resistance was also developed simultaneous with the ATR.

Besides the previous finding that acid tolerance was developed and last for 13 days during the storage at 4 â—¦C for 13 days, the current study investigated the change of cationic antimicrobial peptide (CAMP) resistance during a stimulated storage environment (4 â—¦C for 21 days, the stimulation of storage environment of beef) revealing the potential risk.

On the basis that multiple stress resistances were developed in the current study, we further focus on the role of PhoP/Q two-component system on the transcription regulation of such multiple stress responses. Different from our previous study conducted by Lang et al. (Acid tolerance response of Salmonella during simulated chilled beef storage and its regulatory mechanism based on the PhoP/Q system." Food Microbiol 95: 103716.), in which only genes related to acid tolerance were invaginated, transcriptomics was used to explore the overall role of the PhoP/Q two-component system in regulating multi stress resistance in the current study. Many pathways related to heat, hyperosmotic treatments and specially, cationic antimicrobial peptide (CAMP) resistance was revealed and the antimicrobial peptide (CAMP) resistance pathway was further confirmed on the epigenetic, cell membrane surface charge, and transcriptional levels. This is another innovativeness of our current study.

In above article Salmonella strains isolated from beef processing plants (different serotypes, S. Kottbus, S. Derby, S. Meleagridis, S. Agona, S. Calabar, S. Senftenberg, S. Kingston) were examined. In this study authors examined only 1 reference strain and phoP gene deletion mutation strain (ΔphoP).

Authors previously described and examined S. Kottbus, S. Derby, S. Meleagridis, S. Agona, S. Calabar, S. Senftenberg, S. Kingston isolated from beef processing plants which might be added to this study. Why S. Typhimurium was chosen?

Thanks for the suggestion, different from our previous study mainly focus on the effect of inducing conditions as long as the strain specificity on the development of ATR on Salmonella, the current study mainly focus on if the multiple stress resistances were developed and the underlying mechanisms of such resistances. So only one standard stain and one inducing condition were selected. The chosen of the standard S. Typhimurium can make the comparation of current and following studies easier (such as the measurement of multiple stress resistances and the gene analyzing).

Abstract – a little unclear, spme information about Salmonella, cationic antimicrobial peptide (CAMP), not detailed enough and chaotic. It should be improved.

Response: Thanks for the suggestions, we have described the information of CAMP in more detail in the Abstract section and on revised manuscript line 18, 20-21, 24-26.

The Authors are encouraged to add a schematic illustration, presenting the steps conducted in this study to facilitate the following of the current investigations.

Response: Thanks for the suggestions, we added a schematic diagram of the experimental research ideas as Graphical Abstract.

Line 105 - Acquisition of acid-adapted strains, what control strains were used?

Response: We apologize for the ambiguous expression we used due to inadvertence. The control group of this experiment was the WT strain that was not acid-adapted. We have modified the title to make it easier to understand and on revised manuscript line 103 and 108-112.

Line 111 -… and control strains were obtained by water bath at 37 °C for 90 min for subsequent experiments… - sentence is unclear, please revised

Response: Thanks for the comments, the pre-treatment process of strain acid adaptation was described in more detail to clarify the relationship between the control group and the treatment group and on revised manuscript line 108-112.

Line 113 - 2.3 Determination of acid, heat and salt stress under different conditions of pH and strains - strain obtained in part 2.2 (WT and ∆phoP) and control strains?

2.3 Determination of acid, heat and salt stress under different conditions of pH and strains

Response: Thanks for the comments, the setting of the control group is supplemented in 2.2 of the revised manuscript, and the titles of 2.3 and 2.4 are modified to avoid misunderstanding and on revised manuscript line 113-114 and 139.

Line 123 - 126 – methodology based on reference 28 – where Determination of heat resistance were prepared in TSB not in LB, and according completely different methods.

Response: Thanks for the comments, regarding to the method of the heat resistance experiment, we referred to the reference 28 and made slightly modified by changing the culture medium from TSB to LB. Since the strain was grown in LB during the early growth and acid adaptation process we still used LB during the heat resistance process to ensure the consistency of the strain's living environment), this was mentioned in the revised manuscript, line 125-126.

2.4 Determination of  antibiotic resistance under different conditions of pH and strains

The MICs were interpreted according to the CLSI,  EUCAST or other comitee? It is impotrant especialy for florfenicol. Plase add details. Lack of polymyxin B, peftazidime, gentamycin, florfenicol, ampicillin concentration. Why were the five antibiotics chosen?

Response: Thanks for the suggestions, we interpretation of MICs was based on CLSI M100-S33, and we added the drug concentrations in the revised manuscript as recommended and on revised manuscript line 140-145, 148-153. We also removed florfenicol from the results for the an accuracy expression, because this drug is not a commonly used drug for Gram-negative bacteria.

We chose these antibiotics because they belong to different classes of antibiotics, polymyxin B, ceftazidime, gentamicin and ampicillin were selected to represent antimicrobial peptides, cephalosporin, aminoglycosides and penicillins.

Line 139 - …Less than 5% Dimethyl sulfoxide (DMSO) is added as a co-solvent to antibiotic  solutions that are not readily soluble in water – pleas add this antibiotic names and add  details about study about bacteriostatic effect concentration of DMSO.

Response: Thanks for the suggestions, the antibiotic added with DMSO is ampicillin. The specific details have been stated in the revised manuscript. DMSO is the cosolvent recommended in the CLSI method. References to related research are supplemented in the manuscript as support, and on revised manuscript was reference 27, line 147-148.

How many replicates did you apply for the antibacterial assays? According which guistandards for antimicrobial susceptibility testing MIC methods were performed and why LB broth was used? Most common Medium is Cation-adjusted Mueller-Hinton broth.

Response: Thanks for the comments, we performed three independent replicates of the drug sensitivity test, each with six technical parallels, and this detail has been added in the revised manuscript and on revised manuscript line 150-151.

The culture medium used in the antibiotic sensitivity test was MH medium. We are sorry for the careless expression and have corrected it in the revised manuscript. and on revised manuscript line 143-145.

In paragraph lack of information about antibiotic resistance under different conditions of pH (and strains – please revised sentence). Moreover, cited reference 29 (to line 145-146 or for whole paragraph) described: Effects of clove essential oil and oregano essential oil at sub-MICs on the biofilm formation of S. Derby, but not MIC for antibiotics.

Response: Thanks for the suggestions, the relationship between the treatment of antibiotic broth with different pH and strains during the MIC experiment has been supplemented in the revised manuscript, line 148-153.

In the revised manuscript, reference 29 was replaced with a more relevant reference. The reference number was 28.

2.5 RNA extraction and library sequencing

Line 150 – what is mean: and appropriate  amounts of bacterial precipitates

Response: Thanks for the comments, the amount of bacteria used for RNA extraction was added in the revised manuscript, line 162.

Lack of Quality control strain to MIC methods

Response: Thanks for the suggestions, MIC quality control strain was Escherichia coli ATCC 25922 and the information was added in the revised manuscript on 157-159.

2.9 Determination of drug resistance during storage at low temperature – paragraph is completely unclear. Determination of drug resistance in Cooking of Meat extract (ME)?

Response: We are sorry for the unclear statement. Due to the obvious changes in the sensitivity of Salmonella to CAMP resistance after acid adaptation, we decided to explore the changes in its resistance during long-term storage. Meat extract is used as a culture medium in storage experiments. ME can mimic better the nutritional and environmental conditions experienced by Salmonella in beef storage. We added an explanation to the experimental method to clearly express our research ideas, on the revised manuscript line 228-232.

Results

Line 266 – 268 – concentration of antibiotics should be calryied in methods eg. polymyxin B 4 μg/mL to 32 μg/mL, 4 μg/mL to 2 μg/mL. Lack of origin concentration. Moreover lack of interpretation of MIC results - strains was resistant or susceptible?

Response: Thanks for the comments, the MIC results have been re-described to better understand the changes in MIC and to supplement the CLSI antibiotic susceptibility criteria and on revised manuscript line 280-289 and Table 2; line 404-406 and Table 4.

Figure 4. description of x – axis is completely unreadable

Response: Thanks for the suggestions, Figure 4 was redrawn.

Line 313 – which drug-resistance related pathways genes were downregulated?

Response: Thanks for the comments, transcriptome studies have found that changes in drug resistance pathways in phoP-deficient Salmonella are mainly concentrated in gene expression downregulation, which is related to lipid A modification, cell membrane fluidity, integrity, and porin genes. These gene changes can be summarized as cell membrane modification. Therefore, the drug resistance pathway was removed in the revised manuscript, and the cell membrane modification pathway was retained and on revised manuscript line 330-332.

3.5 Effect of acid adaptation and gene phoP on Salmonella drug resistance during low-temperature storage

What is the link between meat extract (ME) used as an environmental substrate at low temperature and drug resistance?

Response: Thanks for the comments, according to the antibiotic susceptibility testing, acid adaptation will lead to a significant increase in the cationic antimicrobial peptide resistance of Salmonella. The storage experiment is to explore whether the resistance produced by acid adaptation will decrease during the storage beef. The use of ME broth can better simulate the nutritional conditions during meat storage. We added relevant explanations in the method (228-232), and the line of 389-391 in the results also indicates why ME is used.

Lack of reagents producer - non-glucose tryptic soy broth (NTSB), glicerol, lactic acid, petone, NaCl, hydrochloric acid, DMSO, PBS, agarose, probes to remove rRNA, Fragmentation Buffer, M-MuLV reverse transcriptase system, AMPure XP beads, 3-Morpholinopropanesulfonic acid, cytochrome c solution

Response: Thanks for the suggestions, the reagent manufacturer information is added in the revised manuscript, line 101, 102, 108, 116, 117, 118, 146, 161, 166, 169, 171, 173, 174, 216-217, 218-219.

I  have critical points regarding self citation of Xu Gao (1 self citation as first author, 1 as co-author), Jina Han (1 self citation as first author), Lixian Zhu (3 as co-author), George-John E. Nychas (3 as co-author), Yanwei Mao (1 as co-author), Xiaoyin Yang (2 as co-author), Yunge Liu (3 self citation as first author), Yimin Zhang (5 as co-author), Pengcheng Dong (4 as co-author). Authors cited own previus study. It must be changed.

Thanks for the comment. Four of our previous articles were self-cited in the current study. However, three of the four articles had a closely related with the current study and one is cited to demonstrate the method use in the current study. In the study “Liu, Y., et al. (2022). The acid tolerance responses of the Salmonella strains isolated from beef processing plants. Food Microbiology 104”, we confirmed the induce of acid tolerance of Salmonella at different conditions and the risk of ATR in beef processing plants was confirmed. Based on this result, beside the ATR, we further explore the multiple stress resistances using the selected condition conducted by the previous study.

In the study “Han, J. et al. The role of PhoP/PhoQ system in regulating stress adaptation response in Escherichia coli O157:H7. Food Microbiology. 2023, 112, 104244. 10.1016/j.fm.2023.104244” and “Lang, C.X. et al. Acid tolerance response of Salmonella during simulated chilled beef storage and its regulatory mechanism based on the PhoP/Q system. Food Microbiology. 2021, 95”, we mainly disscussed the relationship between the PhoP/Q system and structural genes involved in maintaining acid-base balance metabolism. All of this provided the reader a background information on the regulation role of PhoP/Q system. Base on these information and the method provide, it may make the current easy to understand.

As the reviewer’s comment, we had deleted three self-citation article and keep one to keep the low self-cited rate and also provide information on the source of the ∆phoP strain, the reference number was 25.

Some of the cited articles are outdated. 38 from 61 older than 5 years.

Response: Thanks for the comments, In the revised manuscript, we added some new references to ensure the forward-looking background information of the research, , the reference number was 33, 41 and 48.

Minor comments:

Line 19 - Salmonella Typhimurium instead of Salmonella typhimurium

Response: We are sorry for the carelessness, we corrected the formatting of this strain name, line 19.

Line – 22 ATR - expand abbreviation

Response: We are sorry for the carelessness, the full name of ATR has been completed on line 22.

Line 107, 111 - 37°C instead of 37 °C; Line 109  - 4°C; Line 110, 112, 118 – min.; Line 125, 410 -  55°C.

Response: Thanks for the comments, we have corrected the formatting of the units in the manuscript, line 26, 50, 102, 105, 107, 111, 119, 120, 128, 129, 155, 240, 434, 569.

Line 144 - added antibiotic insted of, added inhibitor,

Response: Thanks for the comments. According to the suggestion, we changed this word in the revised manuscript, line 154.

Round 2

Reviewer 2 Report

Comments and Suggestions for Authors

The Authors have limited themselves to making slight changes. But the main obstacle, which is the way the text is written, has not been modified at all. Therefore, the consideration remains the same.

Authors previously described and examined S. Kottbus, S. Derby, S. Meleagridis, S. Agona, S. Calabar, S. Senftenberg, S. Kingston isolated from beef processing plants which might be added to  this study. It is well established that non-typhoid serovars, particularly those isolated from the environment, exhibit distinct characteristics that differ from those of classical reference strains. 

According the Authors current research is a continuation of previous study, but what about novelty? If above serotypes will be added to study it will be novelty.

Abstract – a little unclear, some information about Salmonella, cationic antimicrobial peptide (CAMP), not detailed enough and chaotic. It should be improved. – not corrected.

The Authors added a schematic diagram of the study as Graphical Abstract, but is completely unreadable.

If control group was the WT strain that was not acid-adapted, it should be added to manuscript.  The Authors modified manuscript line 103 and 108-112 – in manuscript lack of line from 102 to 123

…clarify the relationship between the control group and the treatment group and on revised manuscript line 108-112 - – in manuscript lack of line from 102 to 123

Line 113 - 2.3 Determination of acid, heat and salt stress under different conditions of pH and strains - strain obtained in part 2.2 (WT and ∆phoP) and control strains?

manuscript, was  modified to avoid misunderstanding and on revised manuscript line 113-114 and 139 - in manuscript lack of line from 102 to 123 and line 139 has no changes in text

2.3 Determination of acid, heat and salt stress under different conditions of pH and strains

Methodology based on reference 28 – where Determination of heat resistance were prepared in TSB not in LB, and according completely different methods.

Methods were  modified, but in line 125-126 - has no changes in text

Minimum inhibitory concentration (MIC) was measured by the double dilution  method or according on CLSI M100-S33?

Minimum inhibitory concentration (MIC) was measured by the double dilution  method or according on CLSI M100-S33? Please add origin antibiotic concentration, not stock concentration concentration of 5120 µg/mL. Moreover line 140-145, 148-153 are not in paragraph about MIC

reference 27 is not in line147-148

The Authors performed three independent replicates of the drug sensitivity test – lac in text, line 143-145 are not in paragraph about MIC

Previously cited reference 29 was replaced with a more relevant reference. The reference number was 28, line 148-153 - are not in paragraph about MIC

2.5 RNA extraction and library sequencing

line 162 is not in RNA extraction and library sequencing paragraph

Lack of Quality control strain to MIC methods

Escherichia coli ATCC 25922 was added in the revised manuscript on 157-159 is not in RNA extraction and library sequencing paragraph

2.9 Determination of drug resistance during storage at low temperature

Paragraph is still unclear

Results

Origin concentration of antibiotics should be carried in methods

Figure 4. is still unreadable

What is the link between meat extract (ME) used as an environmental substrate at low temperature and drug resistance?

The Authors added relevant explanations in the method, but in wrong places (228-232), and the line of 389-391 in the results 

Line 20 - an 90 min.  instead of an 90 minutes

Line 25, 252 – Typhimurium instead of typhimurium; 21d – days?

Line 235 - mother liquor?

Whole manuscript – min. – not corrected

Whole manuscript –  it should be unify -  rpm or x g
